# Machine Learning Model-Based Ice Cover Forecasting for a Vital Waterway in Large Lakes

Lian Liu [1], Santhi Davedu [1], Ayumi Fujisaki-Manome [2,3,*], Haoguo Hu [2], Christiane Jablonowski [3] and Philip Y. Chu [4]

1. School for Environment and Sustainability, University of Michigan, Ann Arbor, MI 48105, USA; liulian@umich.edu (L.L.); santhi.davedu@soothsayeranalytics.com (S.D.)
2. Cooperative Institute for Great Lakes Research, University of Michigan, Ann Arbor, MI 48108, USA; hghu@umich.edu
3. Department of Climate and Space Sciences and Engineering, University of Michigan, Ann Arbor, MI 48109, USA; cjablono@umich.edu
4. Great Lakes Environmental Research Laboratory, Office of Oceanic and Atmospheric Research, National Oceanic and Atmospheric Administration, Ann Arbor, MI 48108, USA; philip.chu@noaa.gov
* Correspondence: ayumif@umich.edu; Tel.: +1-734-741-2289

**Abstract:** The St. Marys River is a key waterway that supports the navigation activities in the Laurentian Great Lakes. However, high year-to-year fluctuations in ice conditions pose a challenge to decision making with respect to safe and effective navigation, lock operations, and ice breaking operations. The capability to forecast the ice conditions for the river system can greatly aid such decision making. Small-scale features and complex physics in the river system are difficult to capture by process-based numerical models that are often used for lake-wide applications. In this study, two supervised machine learning methods, the Long Short-Term Memory (LSTM) model and the Extreme Gradient Boost (XGBoost) algorithm are applied to predict the ice coverage on the St. Marys River for short-term (7-day) and sub-seasonal (30-day) time scales. Both models are trained using 25 years of meteorological data and select climate indices. Both models outperform the baseline forecast in the short-term applications, but the models underperform the baseline forecast in the sub-seasonal applications. The model accuracies are high in the stable season, while they are lower in the freezing and melting periods when ice conditions can change rapidly. The errors of the predicted ice-on/ice-off date lie within 2–5 days.

**Keywords:** ice forecast; St. Marys River; Great Lakes; machine learning; LSTM; XGBoost

## 1. Introduction

In the Laurentian Great Lakes (hereafter the Great Lakes), lake ice starts to form in late November and early December [1–3] and causes severe navigation problems in the St. Marys River region. This important waterway connects Lake Superior to Lake Huron and is displayed in Figure 1. In addition, Figure 2 shows a zoomed-in view of the St. Marys River. Federal and commercial icebreaking operations help keep the shipping routes open in early winter and spring, but the navigational lock system is closed from mid-January to late March. In the transition periods (i.e., when lake ice starts to form and melt), an ice forecasting capability with sufficient accuracy and lead time is desired by lock and ice breaking operators as well as the shipping industry.

In any given year, the formation, movement, and timing of the ice cover on the Great Lakes is temperamental and changes substantially with shifts in weather and climate patterns. Extremely cold air across the Great Lakes is a major contributor to ice formation on the Great Lakes. Surface air temperature and its interannual variability are major factors in determining when and how much ice cover develops. These factors are also impacted by teleconnection patterns, such as the El Niño–Southern Oscillation and the North Atlantic

Oscillation. In addition to the large year-to-year fluctuations, the Great Lakes annual maximum ice cover has shown a downward trend since the 1970s [4,5].

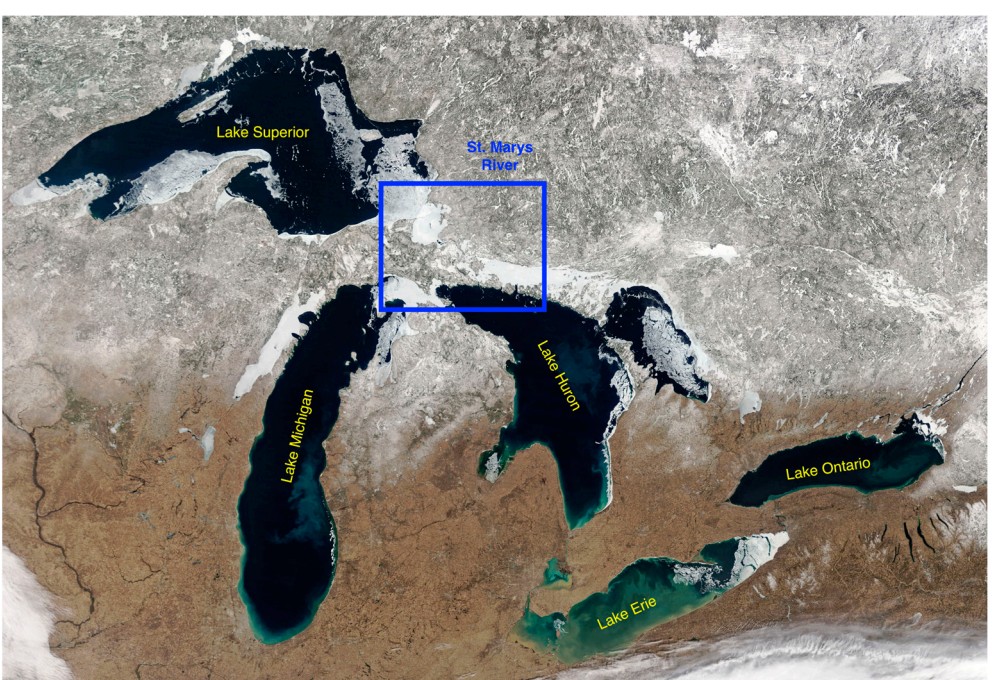

**Figure 1.** A zoomed-in view over the St. Marys River region from Figure 1. St. Marys River's latitude N 46°30.079′ and longitude W 84°25.7424′.

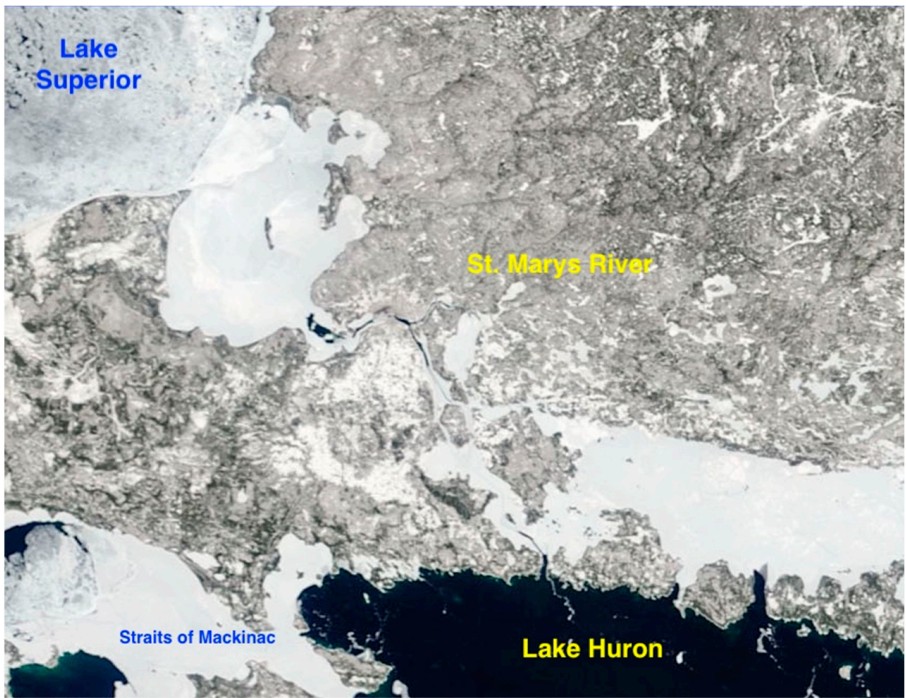

**Figure 2.** Satellite image of the Great Lakes taken on 25 March 2019 based on the Moderate Resolution Imaging Spectroradiometer (MODIS) on NASA's Terra satellite. Adapted from the NASA Earth Observatory website [6].

*1.1. Background on Previous Ice Cover and Analysis Methods*

Several ice cover forecasting studies have been conducted in the Great Lakes area for both short-term (a few days) and seasonal (a few months) time scales [4,7]. They

successfully demonstrated the forecasting capabilities via process-based numerical and statistical models. However, these ice cover forecasting models did not cover the river systems or waterways. This was mainly because it is difficult to capture the detailed, complex physics at these focused geographic scales while also modeling the lake-wide-scale phenomena. As a result, the ice forecasting capability over both short-term and seasonal time scales for the key river systems and waterways is currently limited in the Great Lakes domain, despite the importance of forecast information noted by the shipping community in these domains and the time scales [8].

### 1.2. Overview of Machine Learning Models

Machine learning (ML) techniques have become popular in many scientific fields, such as medicine, finance, geophysics, and climate research [9–12]. They are increasingly used to extract patterns and insights from the ever-increasing volume of geospatial data in the identification of useful connections in the climate system. In addition, they are used to train statistical models which mimic the behavior of climate models. This helps identify and leverage the non-linear relationships between climate variables. However, current approaches may not be optimal if the system behavior is dominated by spatial or temporal patterns. Contextual cues should therefore be used as part of ML (an approach that is able to extract spatio-temporal features automatically) to gain a further understanding of climate science, thus improving the performance of seasonal forecasts, and the modeling of long-range spatial connections across multiple time scales. To date, applications of ML models are limited for the Great Lakes domain, where a few pioneering studies that focused on waves [13,14] and ice/water classification [15]. ML models were also used for ice forecasting tasks in the Arctic Ocean [11,16,17]. However, ML-driven ice forecasting applications have not been developed for the Great Lakes.

ML models, such as the Long Short-Term Memory (LSTM) method and the Extreme Gradient Boost (XGBoost) model are attractive approaches to examining ice forecasting in lieu of numerical modeling. This is especially true for the river systems where lake-wide numerical models are challenged. An obvious advantage of using an ML model versus a numerical geophysical model is the reduced computational cost and possibly better performance for certain processes that numerical models do not necessarily represent well. The Great Lakes waterways fall into this category since they are small but complex systems. Thus, the potential of ML for Great Lakes ice forecasting applications warrants pilot research. This includes finding the most appropriate ML model, identifying the suitable features, and adjusting the best parameters for our prediction problems. Such work is critical in order to support future products that can support decision making by lock operators, vessel managers, ship captains, and U.S. and Canadian Coast Guards around the St. Marys River water system in a way that they can maximize their shipping time and avoid unnecessary cost.

In the present study, we configure two ML models to assess the ice coverage over the St. Marys River, examine various input data including the surface meteorology, water surface temperature, and climate indices, conduct model optimizations, and assess the models' skill for 7-day and 30-day forecast ice predictions.

## 2. Materials and Methods

### 2.1. Model Selection

Our main ML models are the Long Short-Term Memory (LSTM [18]) and Extreme Gradient Boost (XGBoost [19]) techniques. LSTM is a variant of a neural network that is widely used for predicting time series data. XGBoost is a popular ML method that uses a gradient tree boosting technique [14]. Each model has its own learning methods and prediction capabilities, as further described in the following sections.

### 2.1.1. Long Short-Term Memory Networks (LSTM)

Long Short-Term Memory networks are a deep-learning model introduced by Hochreiter and Schmidhuber in 1997 [18]. LSTM is an advanced Recurrent Neural Network (RNN). Compared with RNN, LSTM can store long-term dependency information, which is pivotal for data that have a long-time range. However, RNNs frequently suffer from a problem called vanishing gradient, which causes model learning to become too slow. Consequently, LSTM models are widely applied for time series prediction and have proven to perform better than RNN in these applications.

RNN are always formed by a chain of repeating cells. In a standard RNN, the repeating cells usually have a simple structure such as a "tanh" layer. However, in LSTM, the internal structure is relatively complicated. Instead of a single "tanh" layer, there are four layers interacting in a special way. The structure of an LSTM network is shown in Figure 3.

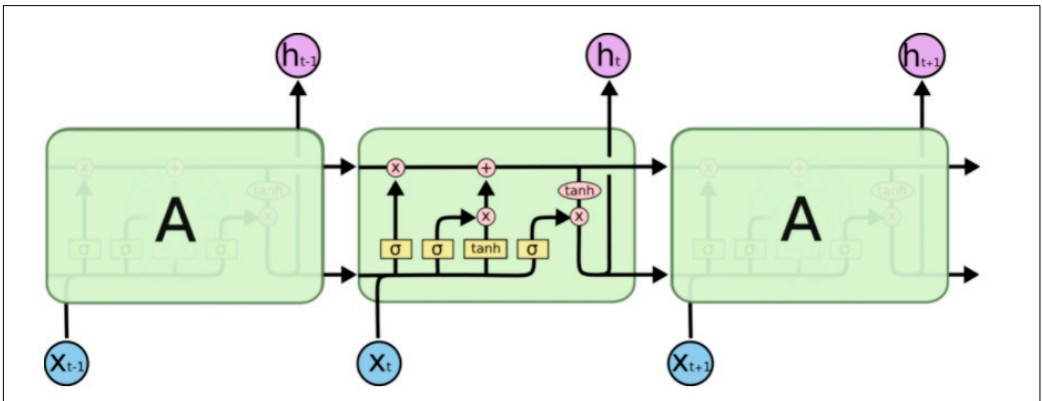

**Figure 3.** The structure of Long Short-Term Memory (LSTM). Adapted from [20].

The principal component of an LSTM is a cell state, which resembles a belt that runs straight down the entire chain. An LSTM can selectively remove or add information to the cell state. Therefore, the LSTM has the ability to solve long-term dependency problems.

In an LSTM model, the cell state is regulated by structures called gates. Gates are a structure that optionally lets information through. Gates are usually composed of a sigmoid neural net layer and a pointwise multiplication operation. It will output a number between zero and one. A zero means that all information will be removed, while a one means all information will get through. There are three types of gates in an LSTM network to control the cell state, which are the forget gate $f_t$ (Equation (1)), the input gate $i_t$ (Equation (2)), and the output gate $o_t$ (Equation (3)). In these three equations, $\sigma$ is a sigmoid function that normalizes the result from 0 and 1. $W$ represents the weighted matrix and b represents the bias vector. In Equation (4), the candidate cell state ($\widetilde{C}_t$) is calculated. Then, the new cell state ($C_t$) is updated based on the result of the forget gate and input gate in equation (5). Finally, in Equation (6), the new output ($h_t$) will be computed based on the new cell state.

$$f_t = \sigma\left(W_f \cdot [h_{t-1},\ X_t] + b_f\right) \tag{1}$$

$$i_t = \sigma(W_i \cdot [h_{t-1},\ X_t] + b_i) \tag{2}$$

$$o_t = \sigma(W_o \cdot [h_{t-1},\ X_t] + b_o) \tag{3}$$

$$\widetilde{C}_t = tanh\ (W_c \cdot [h_{t-1},\ X_t] + b_c) \tag{4}$$

$$C_t = f_t * C_{t-1} + i_t * \widetilde{C}_t \tag{5}$$

$$h_t = o_t * tanh\ (C_t) \tag{6}$$

2.1.2. Extreme Gradient Boost (XGBoost)

XGBoost is a decision-tree-based ensemble ML algorithm that uses a gradient boosting framework [19]. XGBoost is based on decision tree ensembles which are composed of classification and regression trees. Gradient boosting is a supervised machine learning algorithm that attempts to accurately predict a target variable by combining an ensemble of estimates from a set of simpler and weaker models. Boosting is an ensemble learning technique of building many models sequentially where each new model attempts to correct for the deficiencies in the previous model by adding weights. In tree boosting, each new model that is added to the ensemble is a decision tree. Models are added sequentially until no further improvements can be made. The objective of Gradient boosting is to minimize the loss function of the model by aggregating the predictions of weak learners using the gradient descent procedure. Gradient boosting is a greedy algorithm and can overfit the data. Its performance can be improved using tree constraints, shrinkage, random sampling, and penalized learning. XGBoost is an advanced implementation of gradient boosting along with some regularization factors. XGBoost can help reduce overfitting through L1 and L2 regularization by penalizing complex models. It has auto tree pruning where the decision trees do not grow beyond certain limits (specified by the *max_depth* parameter). It has a built-in routine to handle missing values and outliers to a certain extent and has a built-in cross-validation method that runs cross-validation at every iteration of the boosting process. It supports parallel processing at the node level and sequential processing at the tree level. Its capabilities include out-of-core computing, efficient memory management for large datasets, cache optimization and scalability.

*2.2. Data and Preprocessing*

The data used in this study consist of meteorological surface data, water surface temperature data, ice coverage, and climate indices for select teleconnection patterns. The data source and preprocessing procedure for each of the datasets are described in the following sections.

2.2.1. Surface Meteorology

The surface meteorology data were obtained from the Climate Forecast System Reanalysis (CFSR, -2011) and the Climate Forecast System version 2 (CFSv2) operational analysis (2011–2021) [21,22], which were downloaded from the Research Data Archive (RDA). CFSR is a third-generation reanalysis product. It is a global, high-resolution, coupled atmosphere-ocean-land surface-sea ice system designed to provide the best estimate of the state of these coupled domains over this period. CFSv2 provides operational analyses for the years after 2011, which is not covered by CFSR.

The CFSR global atmosphere resolution is ~38 km (T382) with 64 vertical levels. The global ocean has a 0.25° spatial resolution at the equator, extending to a global 0.5° beyond the tropics, with 40 vertical levels. Lake surface temperature and ice coverage in the Great Lakes in CFSR and CFSb2 are based on satellite-based measurements [21,22]. In this research, we mainly focus on five features from the CFSR and CFSv2 datasets, and they are listed in Table 1. All these features are extracted for the domain of the St. Marys River using the latitude and longitude range 46.05–46.59° N and 84.65–83.80° W. The 6-hourly data obtained from the CFSR and CFSv2 datasets were averaged to create daily values.

2.2.2. Water Surface Temperature

Water surface temperature data were downloaded from the National Atmospheric and Oceanic Administration (NOAA) CoastWatch Great Lakes Node website [23]. This dataset contains daily lake-wide-averaged water surface temperature (in Celsius) of Lake Superior, Lake Michigan, Lake Huron, Lake Ontario, Lake Erie, and Lake St. Clair from 1995 to present. In particular, we use water surface temperature data of Lake Superior and Lake Huron since these two lakes are connected to the St. Marys River.

**Table 1.** Surface meteorology features and their units.

| Feature Name | Unit |
|---|---|
| Meridional component of the 10 m wind | meter/second (m/s) |
| Zonal component of the 10 m wind | meter/second (m/s) |
| 2 m air temperature | Kelvin (K) |
| Surface air pressure | Pascal (Pa) |
| Relative humidity | Percentage (%) |

### 2.2.3. Ice Data

Ice data were also downloaded from the NOAA CoastWatch Great Lakes Node [23]. This dataset contains the Great Lakes ice concentration obtained from the U.S. National Ice Center (NIC). The gridded ice analysis products are produced from available data sources including Radarsat-2, Envisat, AVHRR, Geostationary Operational and Environmental Satellites (GOES), and the Moderate Resolution Imaging Spectroradiometer (MODIS). The spatial resolution of the ice concentration data is 2.55 km in 2005, and 1.8 km from 2006 to 2017. The resulting NIC dataset defines ice concentration values from 0 to 100% in 10% steps. As before, the research area was trimmed to the area of St. Marys River by its latitude and longitude (46.05–46.59° N, 84.65–83.80° W). We calculate the average ice concentration over the St. Marys River. A snapshot of this spatial map is shown in Figure 4.

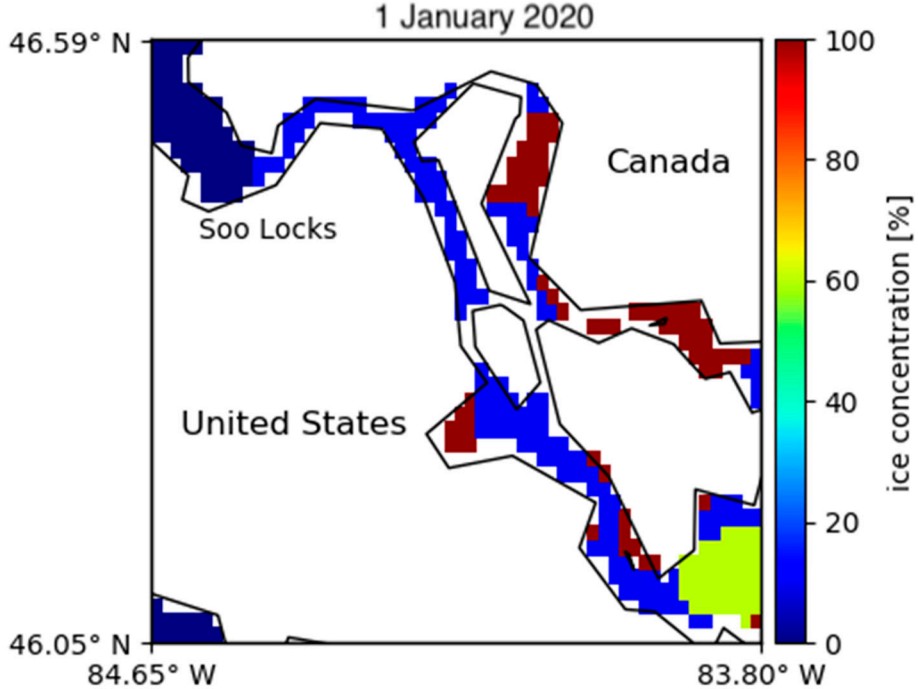

**Figure 4.** Spatial map of average ice concentration of St. Marys River.

### 2.2.4. Climate Indices for Select Teleconnection Patterns

Teleconnection patterns are large-scale patterns of pressure and circulation anomalies that are persistent and recurrent and are spread across vast geographical areas. Climate indices are diagnostic quantities that help to identify and quantify the variability and changes in particular aspects of the climate system [4,24]. Climate indices have become widely used in climate research studies, including those in the Great Lakes [4,24,25]. These indices can help describe the climate in a more approachable way than using the measures of the distribution such as the mean and the standard deviation. Atmospheric teleconnection patterns, especially the North Atlantic Oscillation (NAO) and El Niño, La Niña events were

also found to be associated with mild and severe ice cover on the Great Lakes. In this study, six climate indices are used which are the El Nino–Southern Oscillation (ENSO), North Atlantic Oscillation (NAO), Pacific North/America (PNA), Tropical North Hemisphere (TNH), and Eastern Pacific Oscillation (EPO) or East Pacific/North Pacific Oscillation (EP/NP). The historical data of the climate indices were collected from the National Weather Service Climate Prediction Center and NOAA's Physical Sciences Laboratory websites.

### 2.3. Model Configuration

We have implemented our ML models in Python 3.7.6, with keras-2.3.1 and the tensorflow-2.3.0 backend. Some other relevant libraries include Pandas, Numpy, and Matplotlib. Jupyter Notebook IDE was used for data analysis.

In both the LSTM and XGBoost models, the features specified in Section 2.2. were used as input data and the ice concentration served as the target variable. Both LSTM and XGBoost models predict percent coverage of ice cover over the river. Considering that these are time series data, in which serial correlations exist between successive observations, the usual approach of a random assignment to the three partitions (training, validation, and testing) is not followed. Instead, continuous periods of the time series are assigned to each partition. The data from the previous 25 years, i.e., 1996 to 2021, were used. In our study, an ice season is defined from November in the previous year to the following May. For example, a year "1996" indicates November 1995 to May 1996, covering the winter of 1995/1996. The data were divided into the training set (1996–2010), the validation set (2011–2015), and the test set (2016–2021). With the training set, reliable estimates of the trainable model parameters are achieved. The validation set is used in the selection of model hyperparameters such as the tree depth and to check for overfitting on the training data. Finally, the test set is used to assess how well the ML models generalize to unseen conditions. The model specific configurations are described in Section 2.5. Moreover, the data from November 1 to May 10 are further divided into the freezing phase (1 November–14 January), the stable phase (15 January–25 March) and the melting phase (26 March–10 May). Apart from predicting for the whole winter season (1 November–10 May), we also conducted predictions based on these three phases and observed the performance of our models in different phases. The ML models were trained, validated, and tested for 7-day and 30-day predictions separately.

### 2.4. Evaluation Methods

The following evaluation methods are used to evaluate the accuracy and predictive power of the models. The evaluation metrics are calculated for LSTM, XGBoost and the "baseline" which is the average of all the years of the data considered for the research.

Mean Absolute Error (MAE) and Root Mean Squared Error (RMSE) are generally used to diagnose the variation in errors in the forecasts and evaluate the goodness of fit. MAE is the average of all absolute errors and RMSE is the standard deviation of the residuals. This is represented in Equations (7) and (8), where $N$ is the number of data points, $Xi$ represents original values and $\hat{X}_i$ represents predicted values. Both the MAE and RMSE can range from zero to infinity. They are negatively oriented scores, hence lower values are better. Since the errors are squared before they are averaged, the RMSE is more sensitive to outliers. This means the RMSE is most useful when large errors are particularly undesirable.

$$\text{MAE} = \frac{1}{N} \sum_{i=1}^{n} \left| X_i - \hat{X}_i \right| \tag{7}$$

$$\text{RMSE} = \sqrt{\frac{\sum_{i=1}^{N} \left( X_i - \hat{X}_i \right)^2}{N}} \tag{8}$$

In order to compare the observed and predicted ice-on/off dates, we defined the ice on/off dates as the following: If the ice coverage over the domain (Figure 4) is above 10% for three consecutive days, it will be considered as an ice-on date. If the ice coverage is

below 10% then it is considered an ice-off date. The differences between the observed and predicted ice-on/off dates are calculated as a metric to assess the skills of the two models.

In order to compare the model simulations with the normal year values, we calculate the baseline daily ice coverage time series by calculating 26-year mean values for each day from 1996 to 2021. This baseline provides the approximate information of ice coverage at a given time of a season in the "normal" year. If the error for our model is lower than the baseline, it indicates that our models have better predictive ability than using the normal-year information for the forecast.

### 2.5. Effective Features—Feature Significance

Out of the 12 features (or input variables) in total, we seek to select features that matter most to the target values. In other words, we seek to ensure that every feature we input is meaningful. For LSTM, we calculated the feature importance and implemented a feature selection via the built-in class of the scikit-learn library called "*selectKBest*", which offers a suite of different statistical tests to examine the importance of input variables. The selected regression is called "*f_regression*", which is appropriate for numerical inputs. Features with low importance were removed in the hyperparameters tuning, which is described in Section 2.6.

### 2.6. Hyperparameters Tuning

Hyperparameters were optimized for LSTM using the Sherpa hyperparameter optimization library [26]. For XGBoost, the optimal hyperparameters were chosen manually by evaluating the model performance iteratively. The detailed information on the hyperparameters is described below. The tuning was conducted for the validation set (2011–2015).

#### 2.6.1. LSTM

Hyperparameter tuning for the LSTM model was conducted from four different dimensions. Firstly, we removed the same variables based on the feature selection result. Variables with low feature importance score are removed from the input data one by one to see whether the accuracy can be improved. The test was done both for 7-day prediction and 30-day prediction. We found that the removal of the climate indices SOI, PNA and EPNP maximized the accuracy. Secondly, we found that the "*RMSProp*" optimization method presented the highest accuracy among "RMSProp", "Adam" and "*SGD*". Thirdly, we found that the "*softsign*" activation method presented the highest accuracy among "*relu*", "*elu*", "*softsign*" and "*softmax*". Finally, we explored how the number of hidden neurons and hidden layers and learning rate inside the LSTM affect the prediction accuracy. The accuracy did not have a linear relation with these parameters. Hence, for these three parameters, we used the default learning rate (0.001), 20 hidden neurons, and one hidden layer.

#### 2.6.2. XGBoost

The XGBoost parameters can be divided into three categories. First, the general parameters control the booster type in the model. "booster" and "*nthread*" are examples of such general parameters. Second, the booster parameters control the performance of the selected tree booster. "*max_depth*", "*subsample*", "*colsample_bytree*" are some of the booster parameters. Third, the learning task parameters set and evaluate the learning process of the booster from the given data. "*objective*" and "*eval_metric*" are the learning task parameters. The functioning of each of these parameters as well as the optimized values used in this study are listed in Appendix A. Like LSTM, when the climate indices SOI, PNA and EPNP are removed, the accuracy of the XGBoost model improves slightly.

### 2.7. Shuffling of the Training Data

We conducted an experiment where ice years were shuffled to remove the long-term trend and to add more randomness to the training and validation sets. In this experiment,

we defined an ice year from October 1 to September 30 in the following year (e.g., the 1996 ice year covers 1 October 1995–30 September 1996). This definition preserves the consecutive ice data in a winter and avoids artificial discontinuities in the ice time series that would occur if we defined an ice year based on the calendar year. In this experiment, we shuffled the winter years based on the duration of the winter, i.e., we distributed winters with anomalously long and short durations evenly between the training and validation sets. Table 2 shows how we divided the training set and validation set in this experiment as well as the ice duration for each winter year. We kept the test set the same as before in order to make the comparison between before and after shuffling.

**Table 2.** Ice years in each of the training, validation, and test sets plus the duration of the winter.

| Data | Ice Years and Durations of Winter |
| --- | --- |
| Training set (16 years) | 2012 (94 days), 1998 (101 days), 2010 (113 days), 1999 (114 days), 2007 (129 days), 2002 (134 days), 2006 (135 days), 2001 (138 days), 1997 (139 days), 2008 (139 days), 2009 (141 days), 2013 (146 days), 2015 (160 days), 1996 (163 days), 2014 (165 days) |
| Validation set (6 years) | 2000 (103 days), 2005 (129 days), 2011 (134 days), 2004 (138 days), 1995 (141 days), 2003 (151 days) |
| Test set (6 years) | 2016–2021 |

The period of 1 October–31 December 2015 was not used in the shuffling experiment because it is used in the test set.

## 3. Results and Discussion

We present our prediction results in three ways, which are the time series of the ice coverage, prediction errors, and ice duration. The time series can indicate the variation of the ice concentrations and visually compare the relationship between the observed and predicted values. The prediction errors (RMSE and MAE) of the test set can quantitatively compare the relationship among original values, predicted values and the baseline. Comparisons of the modeled ice duration with observational data can present the error of ice-on/off dates for both models, which is an important metric for the shipping community. Except for the ice duration comparison, the results are only evaluated on the test set (2016–2021).

The results of the feature importance analyses are shown in Figure 5. The ice coverage from the previous days (*Ice1*, *Ice2*, and *Ice3*) were the most important features. The 2 m air temperature three and two days prior to the target date followed (*temperature3*, *temperature2*). Interestingly, the order of the importance for the 2 m air temperature was not in the expected temporal order. This means that the 2 m air temperature three days prior was more important than the 2 m air temperature on the preceding 1–2 days. Next, the water surface temperature in Lake Huron and Lake Superior followed. Surface wind speeds, humidity, and the climate indices were not as important as the above-mentioned features. These features with low importance score were not included in the hyperparameters tuning and final models.

### 3.1. Ice Coverage Time Series

Figure 6 shows the time series of the ice coverage with a 7-day prediction interval. Compared with the green curve (baseline), the temporal variations of the red curve (LSTM) and the blue curve (XGBoost) are more similar to the black curve (observation), which indicates that the model prediction results are more accurate than the baseline.

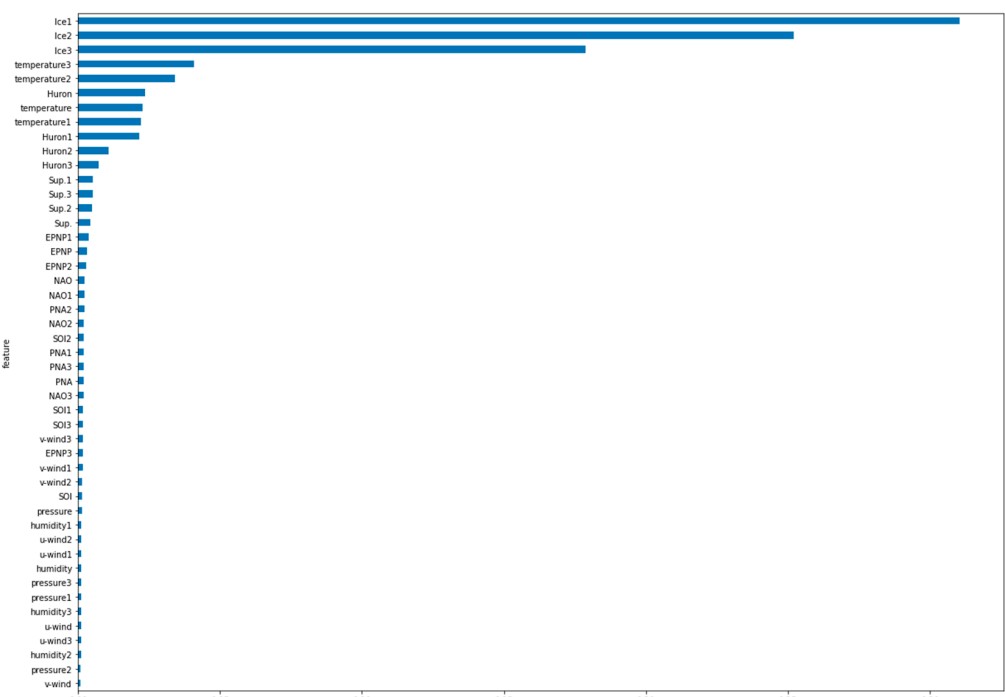

**Figure 5.** Feature significance for XGBoost. A number after a variable name indicates a leading day(s). For example, *Ice1*, *Ice2*, *Ice3* are features that indicate ice coverage over the river system 1 day, 2 days, and 3 days prior, respectively. *temperature* indicates 2 m air temperature. *Huron* and *Sup* indicate lake-wide-averaged water surface temperature in Lake Huron and Lake Superior, respectively.

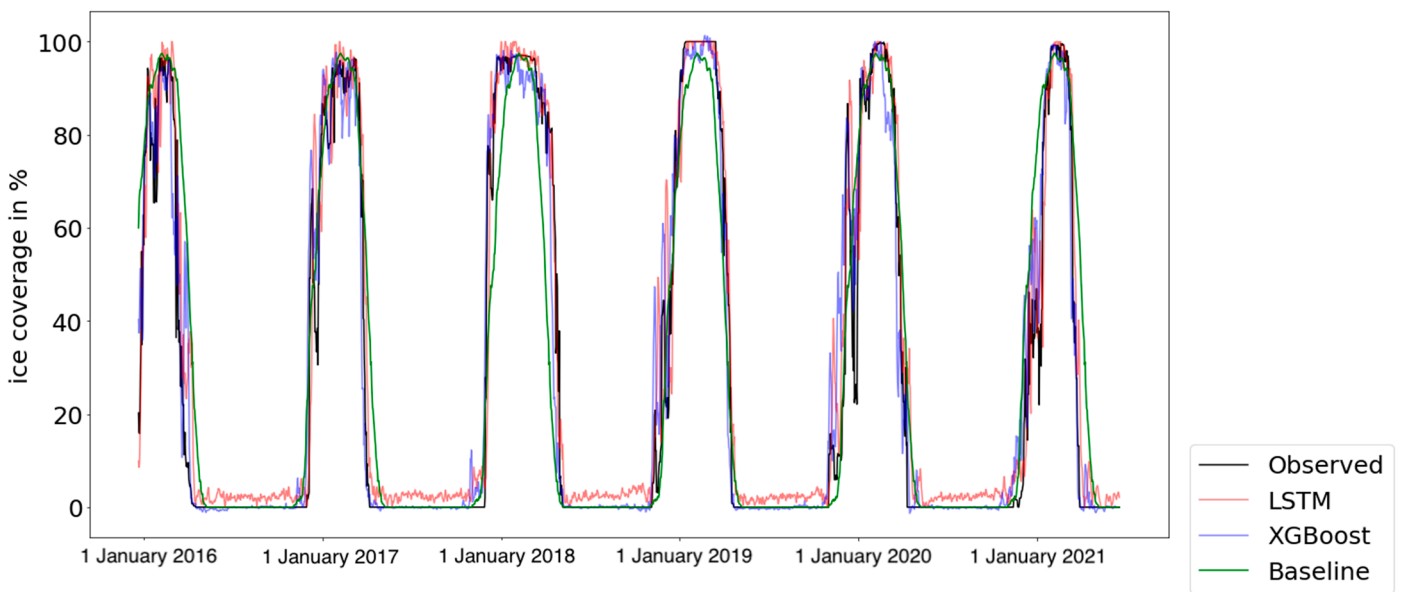

**Figure 6.** Time series of the ice coverage over the St. Marys River system for 7-day predictions.

According to this time series plot, the blue curve (LSTM) and the red curve (XGBoost) overlap with the black curve for most of the time, except in December and April for each year. During these two months, we can see that the black curve (observation) contains some small fluctuations, which indicates that the ice value varies dramatically in a short period. The red curve (LSTM) and the blue curve (XGBoost) both divert from the black curve (observation), indicating that the models have less skills in these two months. The fluctuations in these periods may be caused by a drastic change in the local air temperature,

which is difficult to predict by using the previous weather data, especially when the predicted interval is large.

Table 3 compares the MAE and RMSE among the XGBoost, LSTM and the baseline in the short-term (7-day) and sub-seasonal (30-day) applications. The prediction accuracies of both ML models are better than the baseline when the predicted interval is 7 days. This indicates that the ML models have better accuracy than simply using the normal year values for the forecast.

**Table 3.** Prediction error (RMSE and MAE) on the test set before shuffling.

| Metric | | MAE | | | RMSE | | |
|---|---|---|---|---|---|---|---|
| **Duration** | **Phase** | **XGBoost** | **LSTM** | **Baseline** | **XGBoost** | **LSTM** | **Baseline** |
| 7 day forecast model | Freezing | 10.35% | 10.65% | 11.75% | 15.70% | 15.53% | 17.35% |
| | Stable | 5.92% | 6.04% | 6.3% | 8.92% | 8.81% | 11.37% |
| | Melting | 9.73% | 9.80% | 23.93% | 14.58% | 13.42% | 28.83% |
| | Whole year | 7.64% | 8.75% | 12.72% | 12.43% | 13.09% | 19.21% |
| 30 day forecast model | Freezing | 14.73% | 14.32% | 11.75% | 21.69% | 20.40% | 17.35% |
| | Stable | 8.92% | 9.52% | 6.3% | 13.26% | 13.83% | 11.37% |
| | Melting | 33.55% | 19.32% | 23.93% | 38.91% | 28.09% | 28.83% |
| | Whole year | 33.53% | 13.73% | 12.72% | 38.89% | 20.58% | 19.21% |

For the prediction in the different phases (freezing, stable, and melting), both models present lower RMSEs and MAEs during the stable phase, while they are relatively higher in the freezing and melting phase. This is likely because ice cover tends to change more dramatically in the freezing and melting phases, which makes the prediction difficult.

Figure 7 shows the time series of the ice coverage with a 30-day prediction interval. When compared with the 7-day results, both the red curve and the blue curve have notable differences between the black curve (observation). The green curve (baseline) appears to be closer to the black curve (baseline). The prediction accuracy represented by RMSE and MAE (Table 3) is worse than the baseline for both LSTM and XGBoost. This indicates that our ML models are not well suited for a sub-seasonal prediction.

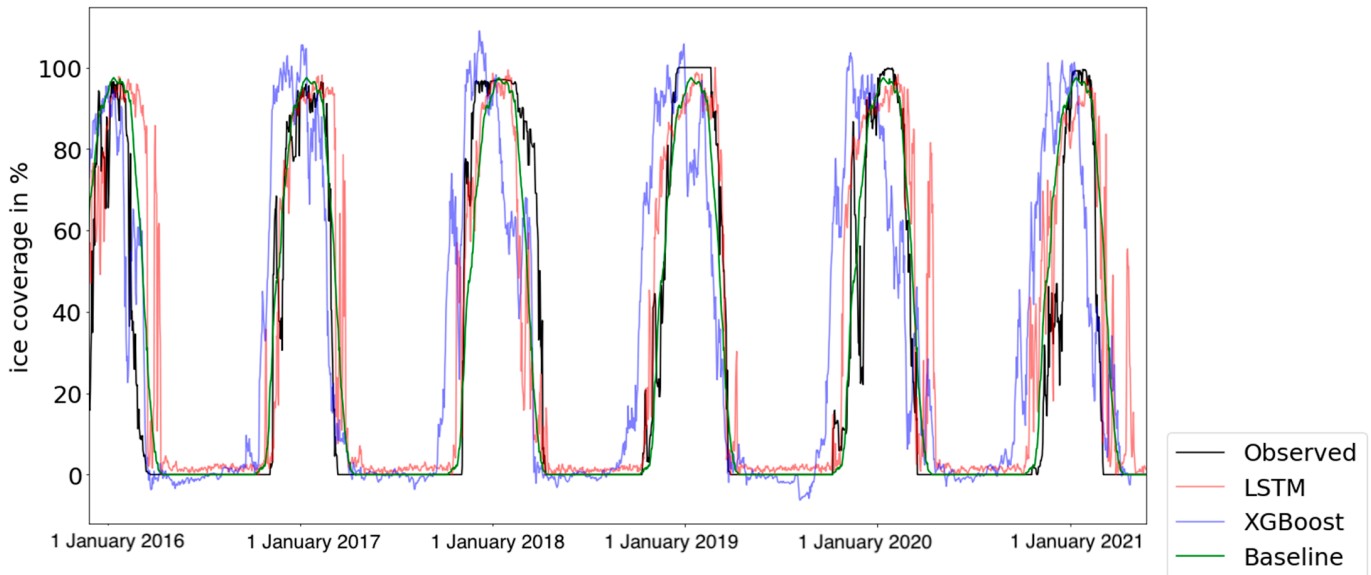

**Figure 7.** Time series of the ice coverage over the St. Marys River system for 30-day prediction.

As for the comparison between LSTM and XGBoost, LSTM performed slightly better than XGBoost did, both for the 7-day and 30-day predictions. In particular for freezing periods and melting periods, the RMSE of XGBoost is notably larger than that of LSTM.

This could be because XGBoost is a tree-based model with input being partitioned into finite possibilities. XGBoost is unable to extrapolate the target values beyond the limits of the training data, e.g., it is limited in predicting outliers that are not captured in the training set. Such situations may occur commonly when ice conditions are rapidly changing. Therefore, XGBoost is less appropriate for predicting ice conditions that are changing drastically. Neural networks such as LSTM generally outperform XGBoost in this regard. On the other hand, XGBoost tends to perform better in the stable period.

### 3.2. Shuffling Ice Years

Table 4 shows the RMSE and MAE from the experiment of the shuffled years for LSTM and XGBoost. For LSTM, the prediction accuracy remains approximately the same after shuffling for the 7-day prediction. For the 30-day prediction, the accuracy becomes even lower than before shuffling, especially for the melting phase. For XGBoost, the prediction accuracy remains unchanged overall. It improves in the stable period after shuffling but tends to be lower in the freezing, melting and whole year periods. Overall, shuffling years did not improve the prediction accuracy significantly for LSTM and XGBoost.

**Table 4.** Prediction error (RMSE and MAE) on the test set after shuffling.

| Metric | | MAE | | | RMSE | | |
|---|---|---|---|---|---|---|---|
| **Duration** | **Phase** | **XGBoost** | **LSTM** | **Baseline** | **XGBoost** | **LSTM** | **Baseline** |
| 7 day forecast model | Freezing | 10.08% | 11.27% | 11.75% | 15.40% | 15.06% | 17.35% |
| | Stable | 5.27% | 4.70% | 6.3% | 7.48% | 7.63% | 11.37% |
| | Melting | 8.26% | 9.26% | 23.93% | 15.73% | 12.73% | 28.83% |
| | Whole year | 8.89% | 8.25% | 12.72% | 14.22% | 12.04% | 19.21% |
| 30 day forecast model | Freezing | 15.21% | 14.98% | 11.75% | 22.76% | 21.35% | 17.35% |
| | Stable | 8.43% | 6.48% | 6.3% | 12.96% | 9.93% | 11.37% |
| | Melting | 34.52% | 29.68% | 23.93% | 39.38% | 38.75% | 28.83% |
| | Whole year | 35.21% | 15.59% | 12.72% | 41.68% | 24.36% | 19.21% |

### 3.3. Ice Season Duration

Figure 8 shows the comparison between the original ice-on/ice-off dates and the predicted ice-on/ice-off dates for 7-day predictions. Since the accuracy of the 30-day prediction was not satisfactory, the ice-on/off plot for the 30-day prediction is not shown here. The black, red and blue lines represent the observed ice duration, and the predicted ice durations of LSTM and XGBoost, respectively.

The differences between the observed and predicted ice-on/off dates for both ML models are small, mostly within 3–5 days. However, for the small ice periods (very short black lines in the plot), both ML models do not capture them accurately.

Overall, the errors of ice-on/ice-off dates were mostly within 5 days for both ML models with the 7-day predicted interval. Interestingly, even in the warmest winters when the lakes were largely ice free (e.g., 1997–1998, 2001–2002, 2005–2006, and 2011–2012 [4]), there were notable ice durations over the river. This is likely because of the shallowness of the river and ice inflow into the river from Whitefish Bay in Lake Superior. This reinforces that forecast models for the lake-wide conditions cannot simply extrapolated to the river system, and therefore it is important to have a forecast capability specific to the river.

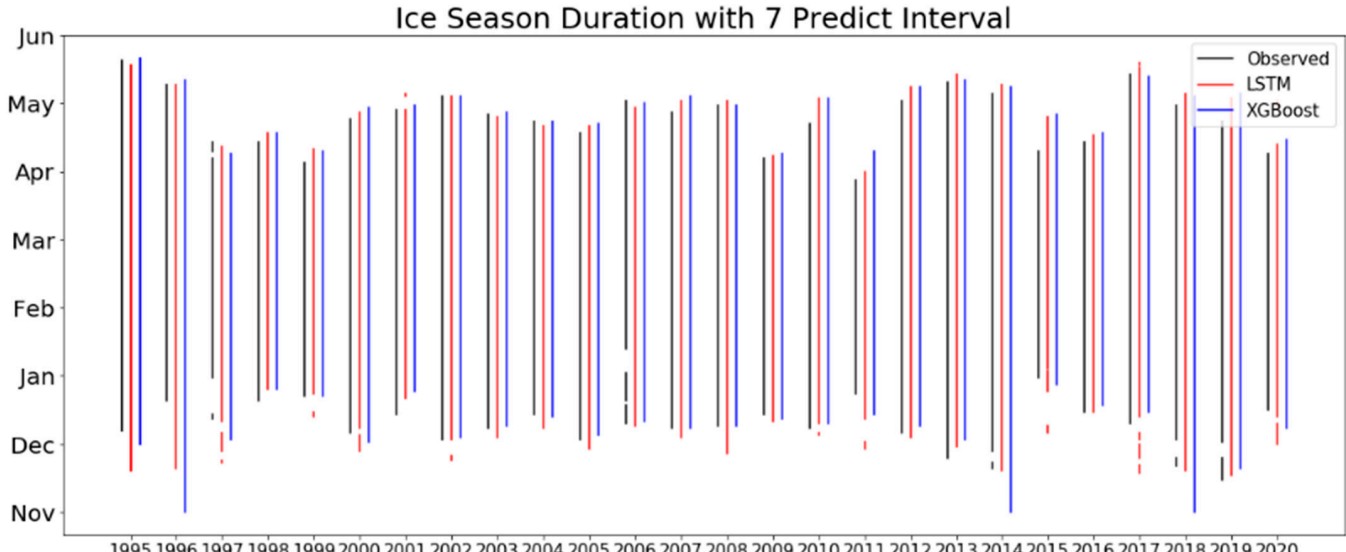

**Figure 8.** Ice duration for 7-day prediction from the winters of 1994/1995–2019/2020.

## 4. Summary and Conclusions

In an effort to develop a decision support tool for waterways and river systems, in this study, we configured, optimized, and evaluated two machine learning models (LSTM and XGBoost) for short-term and sub-seasonal ice prediction with the application to the St. Marys River.

Our analyses showed that both LSTM and XGBoost outperformed the baseline prediction (i.e., normal year prediction) in their short-term predictions (7 days). The prediction error of the ice-on and ice-off dates were within 3–5 days in the short-term predictions. However, the models did not outperform the baseline prediction in the sub-seasonal (30-day) prediction.

For both ML models, the most important features (or input variable) were ice coverage, 2 m air temperature, and lake surface temperature on the previous days. The climate indices, wind speed, and surface pressure were less important in both short-term and sub-seasonal predictions.

Both LSTM and XGBoost performed relatively well in the stable phase (15 January–25 March) compared to the freezing (1 November–14 January) and melting (26 March–10 May) phases. This is likely because the ice conditions in the river system vary notably day by day in the freezing and melting phases, whose dynamic changes are difficult to capture based on the conditions on the previous days. This is likely because XGBoost was not able to extrapolate the target values beyond the limits of training data when making predictions on continuous data.

The limitations of our research are identified as follows: First, machine learning modeling is in general unable to predict any unprecedented event since the model forecasts are based on past data. Second, the gridded atmospheric reanalysis data used in this study do not have sufficient spatial resolution to resolve the river system (i.e., only a few grid boxes over the river region), likely impacting the representation of surface meteorology over the region. Third, the accuracy of the machine learning models is strongly influenced by the predicted interval. Fourth, the ML models in this study do not provide spatial distribution of ice coverage or motion of ice and therefore cannot be a direct replacement of process-based models that can predict dynamic conditions of ice cover in the river system. However, the spatially aggregated time series is still informative to the end users and is appropriate to examine for this pilot study.

With the demonstrated performance in the 7-day forecast applications, this study shows that the two machine learning models provide computationally efficient, verified

alternatives to process-based numerical models for ice forecasting in the St. Marys River and potentially other waterways where ice cover hinders navigational activities.

**Author Contributions:** Conceptualization, A.F.-M.; methodology, validation, and formal analysis, L.L. and S.D.; resources, A.F.-M., H.H., P.Y.C. and C.J.; writing—original draft preparation, L.L and S.D.; writing—review and editing, A.F.-M., C.J. and P.Y.C.; visualization, L.L. and S.D.; supervision, A.F.-M., P.Y.C. and C.J.; project administration, A.F.-M.; funding acquisition, A.F.-M. and C.J. All authors have read and agreed to the published version of the manuscript.

**Funding:** Funding was by the Michigan Data Science Institute 2021 Propelling Original Data Science (PODS) Grants and the National Oceanic and Atmospheric Administration (NOAA) awarded to the Cooperative Institute for Great Lakes Research (CIGLR) through the NOAA Cooperative Agreement with the University of Michigan (NA17OAR4320152). This is a GLERL contribution 2008 and CIGLR contribution 1197.

**Data Availability Statement:** The CFSR and CFSv2 data were downloaded from the Research Data Archive (https://rda.ucar.edu/datasets/ds093.0/, https://rda.ucar.edu/datasets/ds094.0/, access date on 23 July 2022). The ice coverage data and lake surface temperature data were downloaded from the NOAA CoastWatch Great Lakes Node (https://coastwatch.glerl.noaa.gov/, accessed on 23 July 2022). The historical data of the climate indices were collected from the National Weather Service Climate Prediction Center (https://www.cpc.ncep.noaa.gov/, accessed on 23 July 2022) and NOAA's Physical Sciences Laboratory (https://psl.noaa.gov/data/climateindices/list/, accessed on 23 July 2022).

**Acknowledgments:** The authors thank the Michigan Data Science Institute at the University of Michigan for awarding the funding to conduct this study.

**Conflicts of Interest:** The authors declare no conflict of interest. The funders had no role in the design of the study; in the collection, analyses, or interpretation of data; in the writing of the manuscript, or in the decision to publish the results.

## Appendix A. Optimized XGBoost Parameters in This Study

The following settings were used as the optimized XGBoost parameters in the experiment.

1.  General Parameters

    ○ Booster[default = gbtree];
    ○ Booster type = gbtree, gblinear;
    ○ nthread[default = maximum cores available];
    ○ silent[default = 0].

2.  Booster Parameters—Parameters for Tree and Linear Booster

    ○ nrounds = 20;
    ○ learning rate = 0.0005;
    ○ gamma[default = 0];
    ○ max_depth = 3;
    ○ min_child_weight[default = 1];
    ○ subsample = [0.5, 0.6, 0.7];
    ○ colsample_bytree = [0.5, 0.6];
    ○ lambda[default = 0];
    ○ alpha[default = 1].

3.  Learning Task Parameters

    ○ Objective: reg:squared error;
    ○ eval_metric = RMSE—root mean square error (used in regression).

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
