# Peer review of "Machine Learning Model-Based Ice Cover Forecasting for a Vital Waterway in Large Lakes"

_jmse, doi:10.3390/jmse10081022_

Round 1

Reviewer 1 Report

Dear Authors,

Excellent thinking. Any methodology for the prediction of ice levels can surely be helpful for navigation.

An approach I, however, found odd in your methodology was the shuffling of the training data when the aim is to make a time-series forecast: If the target value does actually depend on preceding values, then shuffling the data breaks their relationship. If the target value does not depend on preceding values, then it is arguably not a time-series model, since the order of the observations is irrelevant.

That point aside, the shuffling perhaps also prevented the observation (and learning) of perhaps other trends not directly observable. For instance, has the level of ice or period between the ice-on/off shrank, increased, or stayed the same over the years? Could that be due to global climate change? Or why has it not been effected? Granted the answers to those questions are not the focus of your research but they could be correlations picked up by the LSTM's hidden layers which would then enhance or even validate the predictability of the model.

A second aspect which was not clear to me from the reading of the paper was the needed accuracy for the actionability of the results. How much of an error is acceptable and will not cause the sinking of the Titanic should she decide to set sail based on your LSTM model's forecasts? (Please forgive my poorly imaginative hyperbole there.) I would of course assume this would be of utmost interest to your collaborating and funding institutions?

Nevertheless, the use of LSTMs for forecasting one more time-series is refreshing to see as the usages seem never ending.

Thank you for your research and contribution.

Author Response

Dear Reviewer #1,

Thank you for your time to review this manuscript. We appreciate your inputs and positive comments. Please see the attached response letter that lists point-to-point responses.

Thank you,

Ayumi Manome

Author Response

Dear Reviewer #2,

Thank you for your time to review this manuscript. We appreciate your careful review and constructive feedback, which helped improve the manuscript a lot.  Please see the attached response letter that lists point-to-point responses.

Thank you,

Ayumi Manome
